# Prognostic Significance of Cardiac Magnetic Resonance in Left Atrial and Biventricular Strain Analysis during the Follow-Up of Suspected Myocarditis

**DOI:** 10.3390/jcm12020457

**Published:** 2023-01-06

**Authors:** Yan Chen, Wenjing Zhao, Nan Zhang, Jiayi Liu, Dongting Liu, Zhonghua Sun, Lei Xu, Zhaoying Wen

**Affiliations:** 1Department of Radiology, Beijing Anzhen Hospital, Capital Medical University, Chaoyang District, Beijing 100029, China; 2Discipline of Medical Radiation Science, Curtin Medical School, Curtin University, Perth 6102, Australia

**Keywords:** myocarditis, atrium, ventricle, magnetic resonance imaging, feature tracking, strain

## Abstract

To assess the variation in left atrial (LA) and biventricular strain and its prognostic value in the course of suspected myocarditis, this retrospective study included 55 patients with clinically suspected myocarditis who underwent cardiac magnetic resonance (CMR) examinations at baseline and follow-up periods. Cine images were used for feature tracking analysis. Paired Student’s *t* test, McNemar’s test, and Cox proportional hazard regression were used for statistical analysis. The LA total emptying fraction was the only functional index that showed a statistically significant improvement. The initial LA peak’s late negative strain rate (SRa) was the only parameter with a significant predictive power of major adverse cardiac events under univariable (hazard ratio [HR] 2.396, 95% confidence interval [CI] 1.044–5.498, *p* = 0.039) and multivariable Cox survival analysis when adjusted by LA strain parameters (HR 5.072, 95% CI 1.478–17.404, *p* = 0.010), LA strain and functional parameters (HR 7.197, 95% CI 1.679–30.846, *p* = 0.008), and LA and biventricular strain and functional parameters (HR 10.389, 95% CI 2.250–47.977, *p* = 0.003). Thus, our findings indicate that CMR strain is useful for monitoring LA and ventricular function in suspected myocarditis, that LA function may recover preceding ventricular function changes, and that LA strain may serve as an incremental tool to predict adverse outcomes.

## 1. Introduction

Myocarditis is an inflammatory disease that often occurs in young adults and is the leading cause of cardiac morbidity and mortality in young athletes [1,2]. Athletes with no myocarditis-related symptoms and with normalized serum markers of myocardial injury, inflammation, and heart failure, as well as electrocardiogram results in a retest no less than 3–6 months after initial illness, which refers to “healed myocarditis,” are recommended to return to competitive sports [3]. However, latent inflammation may persist in assumed “healed myocarditis” and result in ventricular tachycardia, dilated cardiomyopathy, and cardiac death, i.e., an adverse prognosis [4,5]. Therefore, reliable indices should be explored to monitor cardiac functional changes and determine parameters that are useful for predicting adverse outcomes in patients with suspected myocarditis during follow-up.

Currently, cardiac magnetic resonance (CMR) is the non-invasive gold standard imaging modality for the diagnosis and follow-up of myocarditis, the diagnostic criteria being referred to as “Lake Louis Criteria” (LLC) [6,7]. Myocardial strain, acquired using the CMR feature tracking (FT) technique, quantifies atrial and ventricular systolic function in different dimensions, with left atrial (LA) and biventricular strains proving to have diagnostic and prognostic significance [8,9]. The prognostic value of ventricular strain has been demonstrated in other cardiac diseases [10,11,12] and myocarditis [13,14,15]. Luetkens et al. [16] explored the alterations in left and right ventricular strain in the course of acute myocarditis and suggested that initial left ventricular (LV) longitudinal strain may serve as a new parameter for the prediction of functional recovery upon follow-up; however, they did not evaluate LA strain in their study. Recently, LA strain has been assessed as a diagnostic and prognostic biomarker for many cardiac disorders [17,18,19]. Dick et al. [20] reported that FT analysis of both atria was feasible in patients with myocarditis, with acceptable reproducibility, and that LA strain parameters showed a good diagnostic value; however, the prognostic value was not reported. Thus, the prognostic value of comprehensive LA and ventricular strain parameters, based on CMR-FT during the follow-up of myocarditis, has rarely been reported. Therefore, this study aimed to assess the alterations in LA and biventricular strain using CMR, as well as its prognostic utility in the course of myocarditis during follow-up.

## 2. Materials and Methods

### 2.1. Patient Population

A total of 79 patients, diagnosed with clinically suspected myocarditis (according to the 2013 ESC guidelines) [1] and who underwent two consecutive CMR examinations between September 2015 and March 2022 at our institution, were included in this study. Clinically suspected myocarditis refers to symptomatic patients presenting with dyspnea, chest pain, palpitations, or other relevant cardiac symptoms who satisfy one or more diagnostic criteria (elevated troponin, abnormal electrocardiography, and functional or structural abnormalities confirmed by echocardiography or CMR) and asymptomatic patients who satisfy two or more of the above-mentioned criteria. Sixteen patients were excluded because of suboptimal images, and eight were excluded because of other cardiac comorbidities (six with ischemic cardiomyopathy, one with hypertrophic cardiomyopathy, and one with cardiac sarcoidosis). Finally, 55 patients (mean age, 15 years; men, 38 [69.10%]) were included in the analysis. This retrospective study was approved by the ethics committee of the authors’ institution, and the requirement for written informed consent was waived.

### 2.2. CMR Protocols

Three different 3.0 T systems, namely, Magnetom Verio (Siemens, Erlangen, Germany), Achieva (Philips, Amsterdam, The Netherlands), and Discovery MR750w (GE Healthcare, Chicago, IL, USA), were used for CMR examinations. Respiratory navigation electrocardiographic gating and parallel imaging techniques were applied. After the initial survey, HASTE black blood sequencing was performed. This was followed by the capture of short-axis T2WI-STIR (black blood T2-weighted imaging short tau inversion recovery) images, as well as cine images of the short-axis and 2-, 3-, and 4-chamber long-axis views. Approximately 10 min after the intravenous injection (at a dose of 0.2 mmol/kg) of gadolinium-based contrast agent (Magnevist, Bayer Healthcare, Berlin, Germany), LGE images of short-axis and 2- and 4-chamber long-axis views were acquired. T1 and T2 mapping images were not acquired. The sequences used in the second CMR examination were the same as those used in the first. The detailed sequence parameters are provided in the Appendix A.

### 2.3. CMR Analysis

Two experienced radiologists with more than 5 years of experience in cardiac medical imaging interpreted the CMR images. Myocardial edema, detected in T2WI-STIR images, and the presence of nonischemic LGE were visually evaluated. Cvi42 software version 5.2.0 (Circle, Calgary, Canada) was used for global LA, LV, and right ventricular (RV) function analysis; volume analysis; and FT analysis. Inter- and intra-observer reproducibility were tested using a random cohort of 14 cases, before FT analysis was performed.

#### 2.3.1. Global LV and RV Function and Volume Analysis

Cine images of the short axis were used, the end-systolic and end-diastolic phases of the LV and RV were automatically defined (they can be the same or different), and the endocardium and epicardium of the LV and the endocardium of the RV in each slice were automatically depicted. The papillary muscle was included in LV volume. The inner and outer rims of the LV and inner rim of the RV were manually adjusted until they could be accurately tracked. Thereafter, indices of the LV and RV function and volume were acquired, including left ventricular end-diastolic volume (LVEDV), left ventricular end-systolic volume (LV-ESV), left ventricular stroke volume (LV-SV), left ventricular ejection fraction (LVEF), LV mass, right ventricular end-diastolic volume (RV-EDV), right ventricular end-systolic volume (RV-ESV), right ventricular stroke volume (RV-SV), and right ventricular ejection fraction (RVEF).

#### 2.3.2. Global LA Function and Volume Analysis

Cine images of 2- and 4-chamber views were used, and the inner rim of the LA (excluding the LA appendage and pulmonary veins) was automatically defined at LV end-diastole, before (LAVpac) and after (LAVmin) LA contraction, as well as at LV end-systole (LAVmax). LA functional indices, including the left total atrial emptying fraction (LAEF), the passive atrial emptying fraction (LAPEF), and the active atrial emptying fraction (LAAEF), were calculated as described in a previous study [21].

#### 2.3.3. FT Analysis

3D global LV and RV and LA myocardial strain analyses are shown in Figure 1. Cine images of the short-axis and 2-, 3-, and 4-chamber views were used for biventricular strain analysis. At the end-diastolic phases (same as those in function and volume analysis) of all cine images, the endocardium and epicardium of the LV were automatically depicted in all four planes, and the endocardium and epicardium of the RV in the short-axis and 4-chamber views were manually depicted. A slight manual adjustment was performed to ensure the accurate tracking of the inner and outer rims of the LV and RV. Thereafter, ventricular strain parameters were obtained, including the global peak strain radial (LV-GRS, RV-GRS), circumferential (LV-GCS, RV-GCS), and longitudinal (LV-GLS, RV-GLS). Two- and four-chamber views of cine images were used for LA strain analysis, with the endocardium and epicardium of the LA manually depicted at LV end-systole; longitudinal curve automatically acquired; and indices of LA strain parameters, including passive strain (εe) and peak early negative strain rate (SRe), active strain (εa) and peak late negative strain rate (SRa), total strain (εs), and peak positive strain rate (SRs), obtained as illustrated previously [21]. εs and SRs correspond to the LA reservoir function, εe and SRe correspond to the LA conduit function, and εa and SRa correspond to the LA contractile booster pump function.

### 2.4. Follow-Up

All patients were followed up through chart reviews or telephone interviews. The primary endpoint was major adverse cardiac events (MACE), including cardiac death, ventricular tachycardia, and rehospitalization due to myocarditis recurrence. The follow-up time was calculated by subtracting the date on which MACE occurred, or the follow-up date, from the date of the second CMR examination.

### 2.5. Statistical Analysis

SPSS (version 25.0; SPSS, Inc., Chicago, IL, USA) was used for statistical analysis. After testing for normal distribution, continuous variables were expressed as mean ± standard deviation or median with interquartile range. Categorical variables were presented as counts and percentages. To evaluate the reproducibility of CMR-FT analysis, inter- and intra-observer agreement was analyzed using intraclass correlation coefficients. Continuous variables were compared using paired Student’s *t* test, and categorical variables were compared using McNemar’s test. Cox proportional hazards regression was used to determine univariable and multivariable associations with MACE, and the results were reported with a 95% confidence interval (CI). A *p* < 0.05 was considered statistically significant.

## 3. Results

### 3.1. Inter- and Intra-Observer Agreement

Inter- and intra-observer agreement assessments of the LA and biventricular strain parameters were performed by two experienced radiologists. To interpret the 14 randomly selected cases, repeated analysis was performed after a 2-week interval. The reproducibility was good for all analyzed strain parameters (see Appendix A).

### 3.2. Study Population and Follow-Up

The median age of the study population was 15 (range: 10–64) years. The baseline clinical characteristics of the study cohort are summarized in Table 1. The time interval between initial symptoms and the first CMR examination was presented as time to CMR1 (median: 19 days), and the time interval between initial symptoms and the second CMR examination was presented as time to CMR2 (median: 202 days). After a median follow-up of 453 (interquartile range: 66–1144) days, MACE occurred in 16 (29.10%) patients, death in 1 (1.82%) patient, ventricular tachycardia in 1 (1.82%) patient, and rehospitalization in 14 (25.45%) patients. These effects were seen due to myocarditis recurrence. Three (5.45%) patients were lost to follow-up.

### 3.3. Alterations in Electrocardiographic, Laboratory, and CMR Index during Follow Up

Differences in electrocardiographic, laboratory, and CMR parameters between the baseline and follow-up are listed in Table 2. The number of patients with abnormal electrocardiograms decreased during follow-up, with a significant difference in ST-T abnormalities, ST elevation, and negative T waves. The levels of serum markers of myocardial injury, inflammation, and heart failure were reduced during follow-up, and differences in the peak levels of high-sensitivity cardiac troponin I and N-terminal pro–B-type natriuretic peptide were significant. The number of patients with visible myocardial edema on T2WI-STIR images and nonischemic myocardial injury on LGE images decreased (Figure 2), as well as the level of those who satisfied the “LLC”; a significant difference was noted in positive T2WI and LGE. LV and LA function recovered, and the difference was only significant in LAEF (58.75 ± 12.59 vs. 61.76 ± 10.21; *p* = 0.033). However, RVEF decreased slightly (47.05 ± 12.75 vs. 46.42 ± 12.04; *p* = 0.745), although the difference was not statistically significant. The absolute value of strain parameters of LV, RV, and LA all increased to a certain extent, and the difference was statistically significant for LVGLS (−10.13 [−12.06, −6.98] vs. −10.64 [−13.20, −8.98]; *p* = 0.005), RVGRS (28.92 [21.70, 39.74] vs. 33.92 [21.16, 49.53]; *p* = 0.019), εs (20.70 [12.90, 25.90] vs. 25.00 [16.90, 34.80]; *p* = 0.001), εe (11.80 [7.30, 18.30] vs. 15.30 [11.30, 23.00]; *p* = 0.002), and SRa (−1.10 [−1.55, −0.80] vs. −1.40 [−1.80, −1.00]; *p* = 0.001; Figure 3).

### 3.4. Prognostic Value of Strain Parameters during Follow-Up

When all the second laboratory and CMR parameters were subjected to univariable Cox proportional hazards regression analysis, none of them showed a significant association with MACE (see Appendix A). The prognostic values of initial laboratory and CMR parameters were calculated. In the univariate analysis (Table 3), only the LA SRa was a significant outcome predictor (hazard ratio [HR] 2.396, 95% CI 1.044–5.498; *p* = 0.039). In the multivariable model for MACE (Table 4), SRa also showed significant prognostic value when adjusted by LA strain parameters (including εs, εa, εe, SRe, and SRs; HR 5.072, 95% CI 1.478–17.404, *p* = 0.010); LA strain and functional parameters (including εs, εa, εe, SRe, SRa, LAAEF, LAPEF, and LAPEF; HR 7.197, 95% CI 1.679–30.846; *p* = 0.008); and all LA, LV, and RV strain and functional parameters (including εs, εa, εe, SRe, SRa, LAAEF, LAPEF, LAPEF, LVEF, RVEF, LVGRS, LVGCS, LVGLS, RVGRS, RVGCS, and RVGLSd; HR 10.389, 95% CI 2.250–47.977; *p* = 0.003).

## 4. Discussion

CMR plays an important role in the follow-up of patients with myocarditis. Our study included patients with clinically suspected myocarditis who underwent two CMR examinations. This is the first study to comprehensively evaluate the alterations in LA and biventricular strain and the relationship between strain parameters and MACE during follow-up. The key findings of this study are as follows: (1) LA and biventricular myocardial strain improved upon follow-up, and LAEF was the only significantly recovered functional parameter; and (2) the initial LA SRa was the only significant predictor of MACE upon follow-up.

### 4.1. Alteration of Electrocardiographic, Laboratory, and CMR Index during Follow-Up

During the follow-up period, acute myocarditis progressed into the chronic phase; the electrocardiogram tended to normalize; the levels of serum markers of myocardial injury, inflammation, and heart failure were reduced; and the proportion of patients with visible myocardial edema and myocardial injury decreased. Over the course of myocarditis, as reported earlier by Luetkens et al. [16], LV and RV function and myocardial strain significantly improved on follow-up CMR. In their recent study, Sperlongano et al. [22] reported that LV function and GLS, both based on echocardiography, significantly improved from baseline to follow-up. Our study not only demonstrated the decrease in inflammatory biomarker levels, revealed improved LV function, as well as LV and RV myocardial strain, as in these previous studies, but also reported improved LA function and strain.

LA function in cardiac disorders has attracted wide attention in recent years, and its dysfunction plays an important role in cardiovascular diseases, including atrial fibrillation, cardiomyopathies, heart failure, and valvular heart diseases [23,24]. However, LA function assessment in myocarditis has only been reported in a few studies. Doerner et al. [25] found that LA εe and SRe were significantly reduced in patients with myocarditis compared with healthy controls and reported impaired LA function in myocarditis; however, they failed to explore the role of this condition in the recovery course. In our study, LAEF was identified as the only functional parameter showing statistically significant recovery, whereas the improvement in LV function was not statistically significant. This finding may be the result of the involvement of LA in the inflammatory process of myocarditis, as demonstrated recently in a murine model with viral myocarditis [26]. However, it may also be due to a change in LV filling pressures, which was believed to be a determinant of the LA reservoir and pump strain that were measured by speckle tracking echocardiography [27]. It was also indicated that LA function may be more sensitive to early changes than LV function and may recover prior to LV and RV function changes in myocarditis follow-up, which may add some information to the prognostic value of LA strain in myocarditis. The implication of this insignificant decrease in RVEF requires further investigation.

### 4.2. Prognostic Value of Strain Parameters during Follow-Up

In our study, LA strain improvement was statistically significant, and the initial LA SRa was the only significant predictor of MACE, while the second LA strain upon follow-up was not associated with poor patient outcomes. The prognostic value of LA strain has been validated in the context of other cardiovascular diseases. Zhou et al. found that LA strain was more sensitive than LV GLS in evaluating the primary endpoint in hypertrophic cardiomyopathy and hypertension [24]. A study by Chirinos et al. found that in patients with heart failure, all measures of phasic LA function and volume were associated with the composite outcome, whereas the conduit and reservoir function of LA were significantly correlated with the composite outcome. This was even the case in analyses that adjusted for clinical risk factors, HF status, maximum LA volume, LV mass, and LVEF [28]. Although some studies have demonstrated the impairment and diagnostic value of LA strain in myocarditis, as stated previously [20,25], this study is the first to explore the prognostic value of LA strain during the course of myocarditis. Upon follow-up, none of the LV and RV strain parameters showed a significant association with MACE; initial LA SRa was the only independent outcome predictor, even when adjusted by atrial and biventricular functional and strain parameters. This novel finding may indicate that initial LA function is a strong outcome predictor upon follow-up, is incremental to LV and RV, and may serve as a useful tool to guide long-term therapy.

### 4.3. Limitations

Our study has some limitations. First, the enrolled patients were diagnosed with clinically suspected myocarditis, “LLC” was not satisfied in all patients; parametric imaging, including mapping and assessments of extracellular volume (ECV), was absent; no patients received endomyocardial biopsy (EMB) confirmation. Second, the sample size of the study was relatively small. Further studies with larger sample size are needed to verify our findings.

## 5. Conclusions

During the course of suspected myocarditis, CMR strain analysis is useful for monitoring both atrial and ventricular function, and LA function may recover before LV and RV function changes. Initial LA strain may serve as a useful tool to predict adverse outcomes during follow-up, and may be incremental to ventricular strain parameters.

## Figures and Tables

**Figure 1 jcm-12-00457-f001:**
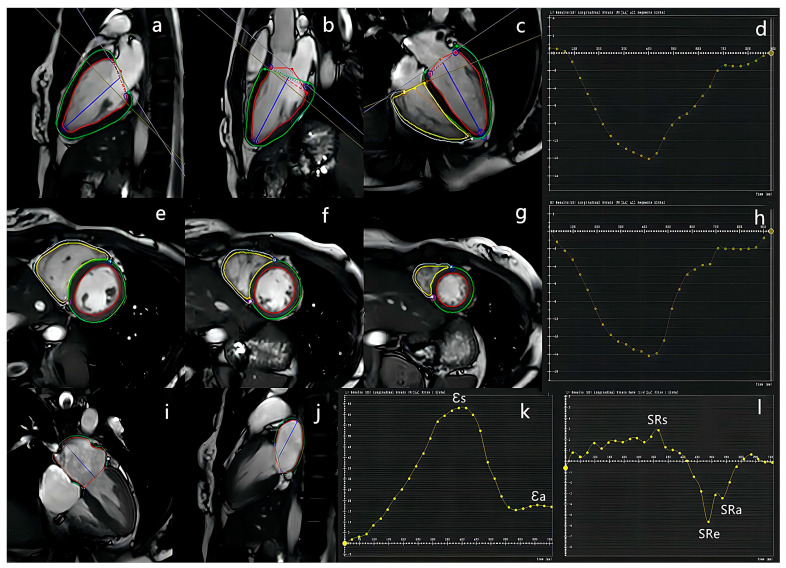
Representative example of left atrial and biventricular strain analysis. 2-, 3-, and 4-chamber view long-axis (**a**–**c**) and short-axis ((**e**–**g**), from bottom to apex) cine images were used for ventricular strain analysis. The endo- and epicardial borders of both ventricles were depicted, red and green lines in picture (**a**–**c,e**–**g**) refer to LV endo- and epicardial borders, respectively; yellow and blue lines in picture (**c**,**e**–**g**) refer to RV endo- and epicardial borders, respectively; and the curves of peak global longitudinal of LV (**d**) and RV (**h**) were automatically constructed. 2- and 4-chamber view long-axis (**i,j**) cine images were used for left atrial strain analysis. The endo- (red lines) and epicardial (green lines) borders of the LA were manually depicted, and the curves of peak strain (**k**) and strain rate (**l**) of the LA were automatically constructed.

**Figure 2 jcm-12-00457-f002:**
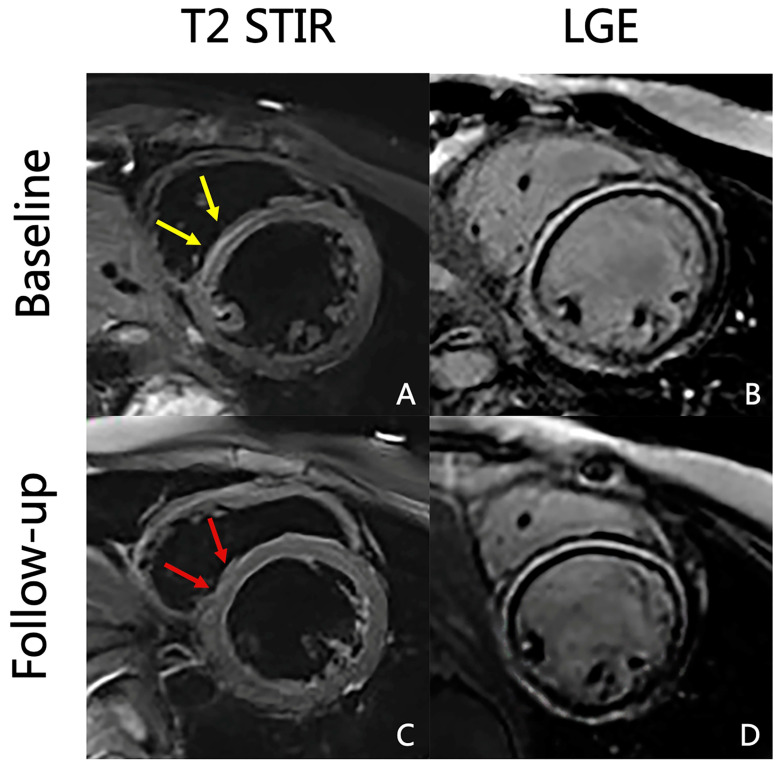
Baseline and 3-month follow-up T2 STIR and LGE images of a 15-year-old man with acute myocarditis. Myocardial edema in the basal ventricular segment of septal wall resolved from baseline ((**A**), yellow arrows) to follow-up ((**C**), red arrows). However, the LGE persisted (**B**,**D**).

**Figure 3 jcm-12-00457-f003:**
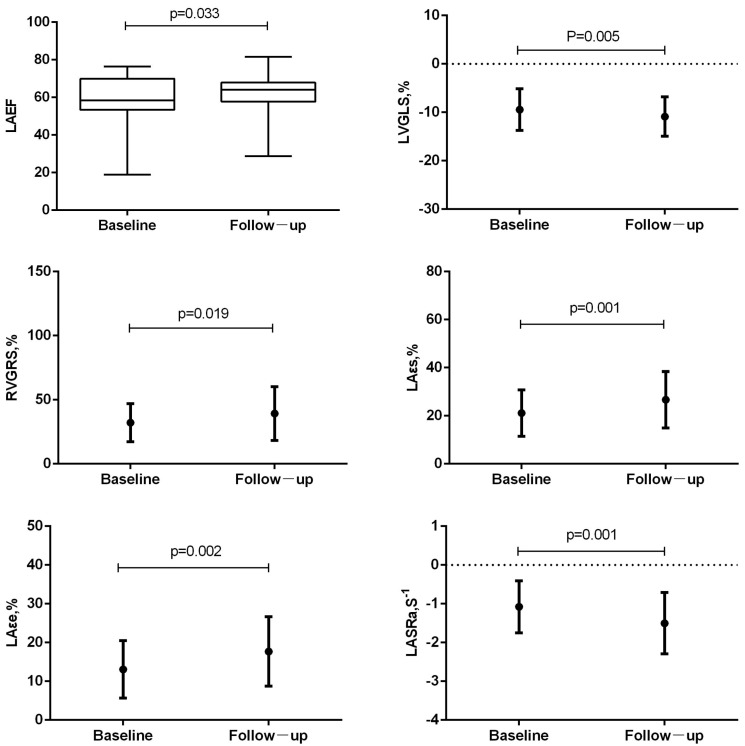
Differences in the left atrial and biventricular function and myocardial strain between baseline and follow-up period. LAEF was the only significantly recovered functional parameter. The absolute value of LVGLS, RVGRS, LA εs, εe, and SRa was significantly increased.

**Table 1 jcm-12-00457-t001:** Clinical characteristics of patients at baseline.

Parameter	Baseline (*n* = 55)
Age (years)	15 (13, 23)
Male patients (%)	38 (69.10)
Body mass index (kg/m^2^)	22.44 ± 5.14
SBP (mmHg)	114.33 ± 14.70
DBP (mmHg)	68.82 ± 9.43
HR	84.65 ± 17.23
Time to CMR1 (days)	19 (7, 53)
Time to CMR2 (days)	202 (146, 292)

SBP: systolic blood pressure; DBP: diastolic blood pressure; HR: heart rate; CMR: cardiac magnetic resonance.

**Table 2 jcm-12-00457-t002:** Laboratory and CMR parameters of patients at baseline and follow up.

Parameter	Baseline (*n* = 55)	Follow up (*n* = 55)	*p* Value
Abnormal ECG			
Low voltage	2 (3.64)	1 (1.82)	0.564
ST-T abnormalities	33 (60.00)	14 (25.45)	0.001
ST elevation	30 (54.55)	10 (10.18)	<0.001
Negative T	14 (25.45)	1 (1.82)	<0.001
Significant Q-wave	3 (5.45)	0 (0.00)	0.250
Laboratory parameters			
Peak level CK (U/L)	85 (65, 153)	86 (54, 130)	0.145
Peak level hs-TNI (ng/mL)	0.396 (0.00, 2.60)	0.00 (0.00, 0.80)	0.012
Peak level NT-proBNP (pg/mL)	32.30 (13.60, 73.80)	22.50 (12.40, 62.50)	0.018
Peak level CRP (mg/L)	0.89 (0.21, 5.57)	0.65 (0.08, 3.62)	0.143
WBC count (×10^9^/L)	7.00 (5.92, 8.12)	6.64 (5.44, 8.12)	0.305
Traditional CMR features			
Positive T2WI	21 (38.18)	10 (18.18)	0.007
Positive LGE	28 (50.91)	21 (38.18)	0.016
Positive “LLC”	10 (18.18)	5 (9.09)	0.063
LVEDV, mL	143.19 ± 55.86	143.82 ± 56.03	0.876
LVESV, mL	59.79 (38.71, 72.16)	55.64 (41.71, 55.64)	0.241
LVSV, mL	76.46 ± 25.41	81.41 ± 24.80	0.109
LVEF, %	56.33 ± 13.99	58.43 ± 10.19	0.116
LV mass, g	95.12 ± 34.90	91.93 ± 33.76	0.141
RVEDV, mL	137.23 ± 39.36	140.71 ± 41.53	0.381
RVESV, mL	69.70 (53.87, 88.37)	74.00 (51.58, 96.53)	0.209
RVSV, mL	63.66 ± 24.79	64.86 ± 26.03	0.732
RVEF, %	47.05 ± 12.75	46.42 ± 12.04	0.745
LAVmax, mL	46.90 (36.86, 60.86)	48.40 (38.79, 62.94)	0.657
LAVmin, mL	17.78 (13.79, 23.10)	18.06 (13.22, 23.83)	0.535
LAVpac, mL	39.76 (31.97, 52.48)	41.22 (31.36, 53.46)	0.575
LAEF, %	58.75 ± 12.59	61.76 ± 10.21	0.033
LAPEF, %	14.31 (7.94, 21.11)	14.19 (7.39, 21.55)	0.980
LAAEF, %	16.70 (8.62, 26.77)	16.54 (7.98, 27.47)	0.874
Feature tracking			
Global ventricular peak strain			
LVGRS, %	28.94 ± 12.88	30.88 ± 10.21	0.313
LVGCS, %	−17.75 ± 5.27	−19.07 ± 3.56	0.057
LVGLS, %	−10.13 (−12.06, −6.98)	−10.64 (−13.20, −8.98)	0.005
RVGRS, %	28.92 (21.70, 39.74)	33.92 (21.16, 49.53)	0.019
RVGCS, %	−2.80 (−8.72, 11.26)	−5.83 (−8.95, 8.78)	0.137
RVGLS, %	−7.50 (−11.27, −5.28)	−9.44 (−12.42, −6.50)	0.144
LA strain			
εs, %	20.70 (12.90, 25.90)	25.00 (16.90, 34.80)	0.001
εa, %	7.40 (4.40, 11.90)	7.50 (4.50, 11.70)	0.311
εe, %	11.80 (7.30, 18.30)	15.30 (11.30, 23.00)	0.002
SRs, s^−1^	1.40 (1.00, 1.80)	1.40 (1.10, 1.90)	0.176
SRe, s^−1^	−1.90 (−2.60, −1.30)	−2.00 (−2.80, −1.25)	0.479
SRa, s^−1^	−1.10 (−1.55, −0.80)	−1.40 (−1.80, −1.00)	0.001

ECG: electrocardiogram; CK: creatine kinase; hs-cTNI: high-sensitivity cardiac troponin I; NT-proBNP: N-terminal pro–B-type natriuretic peptide; CRP: c-reactive protein; WBC: white blood cell; CMR: cardiac magnetic resonance; T2WI: T2-weighted imaging; LGE: late gadolinium enhancement; “LLC”: “Lake Louis Criteria”; LV: left ventricle; RV: right ventricle; ESV: end systolic volume; EDV: end diastolic volume; SV: stroke volume; EF: ejection fraction; LAVmax: maximal left atrial volume; LAVmin: minimal left atrial volume; LAVpac: pre-atrial contraction left atrial volume; LAEF: total left atrial emptying fraction (EF total); LAPEF: passive left atrial emptying fraction (EF passive); LAAEF: active left atrial emptying fraction (EF booster); GRS: global radial strain; GCS: global circumferential strain; GLS: global longitudinal strain; LA: left atrium; εs: total strain; εa: active strain; εe: passive strain; SRs: peak positive strain rate; SRe: peak early negative strain rate; SRa: peak late negative strain rate.

**Table 3 jcm-12-00457-t003:** Univariable association of initial laboratory findings and CMR parameters with MACE.

Parameter	HR (95% CI)	*p* Value
Peak level CKda (U/L)	0.995 (0.988, 1.003)	0.222
Peak level hs-TNId (ng/mL)	0.996 (0.969, 1.024)	0.774
Peak level NT-proBNPd (pg/mL)	1.000 (1.000, 1.000)	0.812
Peak level CRPd (mg/L)	1.020 (0.948, 1.096)	0.600
WBCd (×10^9^/L)	0.919 (0.747, 1.130)	0.422
Positive T2WI	0.614 (0.213, 1.772)	0.367
Positive LGE	0.515 (0.187, 1.422)	0.201
Positive “LLC”	0.668 (0.152, 2.943)	0.594
LVEF, %	0.982 (0.938, 1.029)	0.450
RVEF, %	0.970 (0.924, 1.019)	0.227
LAEF, %	0.996 (0.955, 1.038)	0.831
LAPEF, %	1.048 (0.993, 1.106)	0.089
LAAEF, %	1.032 (0.999, 1.066)	0.060
LVGRS, %	0.989 (0.950, 1.029)	0.576
LVGCS, %	1.087 (0.987, 1.196)	0.089
LVGLS, %	1.042 (0.944, 1.149)	0.416
RVGRS, %	1.014 (0.983, 1.046)	0.387
RVGCS, %	1.015 (0.979, 1.053)	0.421
RVGLS, %	1.008 (0.967, 1.051)	0.710
εs, %	0.976 (0.925, 1.030)	0.382
εa, %	1.036 (0.929, 1.155)	0.526
εe, %	0.944 (0.876, 1.017)	0.132
SRs, s^−1^	0.729 (0.314, 1.690)	0.461
SRe, s^−1^	1.494 (0.851, 2.621)	0.162
SRa, s^−1^	2.396 (1.044, 5.498)	0.039

HR: hazard ratio; CI: confidence interval; other abbreviations same as in Table 2.

**Table 4 jcm-12-00457-t004:** Multivariable Cox regression analysis with initial SRa for MACE.

SRa	HR (95% CI)	*p* Value
Model 1	5.072 (1.478, 17.404)	0.010
Model 2	7.197 (1.679, 30.846)	0.008
Model 3	10.389 (2.250, 47.977)	0.003

Model 1 adjusted for: εsd, εad, εed, SRed, and SRad. Model 2 adjusted for: εsd, εad, εed, SRed, SRad, LAAEFd, LAPEFd, and LAPEFd. Model 3 adjusted for: εsd, εad, εed, SRed, SRad, LAAEFd, LAPEFd, LAPEFd, LVEFd, RVEFd, LVGRSd, LVGCSd, LVGLSd, RVGRSd, RVGCSd, and RVGLSd. Abbreviations as in Table 2 and Table 3.

## Data Availability

Not applicable.

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
