# Peer review of "Prognostic Significance of Cardiac Magnetic Resonance in Left Atrial and Biventricular Strain Analysis during the Follow-Up of Suspected Myocarditis"

_jcm, 2023, doi:10.3390/jcm12020457_

Round 1
Reviewer 1 Report
Dear Authors,
in your paper entitled 'Prognostic Significance of Cardiac Magnetic Resonance In the 2 Analysis of Left Atrial and Bi-ventricular Strain during Follow- 3 up of Acute Myocarditis' the aim of the study is to assess the variation of left atrial (LA) and bi-ventricular strain and its prognostic value in the course of acute myocarditis. methods are well described, but the follow-up time is not clear. Authors demonstrated that LA strain improvement showed statistical significance between the second and the first examinations and it was the only significant predictor of MACE upon follow-up. This is a nice finding but in clinical practice we can perform LA strain also with echocardiography that is a more diffuse and simply technique; CMR is not always available and has long time of exposure.
Furthermore figure 2 panel A is not rapresentaative of edema (the white line is the blood pool) and the decription of the figure is not correct.
Kind Regards
Author Response
in your paper entitled 'Prognostic Significance of Cardiac Magnetic Resonance In the 2 Analysis of Left Atrial and Bi-ventricular Strain during Follow- 3 up of Acute Myocarditis' the aim of the study is to assess the variation of left atrial (LA) and bi-ventricular strain and its prognostic value in the course of acute myocarditis. methods are well described, but the follow-up time is not clear. Authors demonstrated that LA strain improvement showed statistical significance between the second and the first examinations and it was the only significant predictor of MACE upon follow-up. This is a nice finding but in clinical practice we can perform LA strain also with echocardiography that is a more diffuse and simply technique; CMR is not always available and has long time of exposure.
Furthermore figure 2 panel A is not rapresentaative of edema (the white line is the blood pool) and the decription of the figure is not correct.
Response: Thank you for your comments. As suggested, we replaced Figure 2 with another patient with obvious edema on T2WI.
Reviewer 2 Report
This study aimed to evaluate CMR quantified Left Atrial and Bi-ventricular Strain during Follow-up of Acute Myocarditis and its prognostic significance.
The study is of clinical importance. However, there are a few issues some problems in this manuscript:
Q1:
Alterations in LV, and RV functional strain parameters frequently occur during the acute stage of myocarditis. During the course of the disease, a significant improvement in LV and RV strain parameters can be observed. Initial LV longitudinal strain may serve as a new parameter for predicting functional recovery upon follow-up. it is a little weird that no significance was proved in this work. Authors need to mention such facts.
Q2: I think it would be better to compare each patient with him/herself. Furthermore, Data collection is very interesting (median age 15). Authors should reanalyze by rolling out older patients.
Q2:
It seems that the reduction in the statistical significance can be improved when switching to a narrowed heart rate range. It further seems that there can be many other ways of investigating the relationship between heart rate and strains. For example, IDI, NRI, AIC or BIC analysis comparing models where heart rate is included and excluded, evaluation of correlation strengths of individual parameters in modelling, leave-one-out analysis etc. The authors are urged to consider this.
Weise Valdés et al, Left-Ventricular Reference Myocardial Strain Assessed by Cardiovascular Magnetic Resonance Feature Tracking and fSENC-Impact of Temporal Resolution and Cardiac Muscle Mass
November 2021Frontiers in Cardiovascular Medicine 8:764496
Q3: The methodology is acceptable, but the absence of mapping and ecv should be mentioned in the limitation.
Author Response
Q1:
Alterations in LV, and RV functional strain parameters frequently occur during the acute stage of myocarditis. During the course of the disease, a significant improvement in LV and RV strain parameters can be observed. Initial LV longitudinal strain may serve as a new parameter for predicting functional recovery upon follow-up. it is a little weird that no significance was proved in this work. Authors need to mention such facts.
Response: Thank you for your comments. Univariate association of initial laboratory and CMR parameters with MACE was added in the manuscript to replace the previous Table 3. Initial LV and RV strain parameters were not significantly associated with MACE, while initial LA SRa showed significant association with MACE.
Q2: I think it would be better to compare each patient with him/herself. Furthermore, Data collection is very interesting (median age 15). Authors should reanalyze by rolling out older patients.
Response: Thanks for the constructive advice. Our study retrospectively included all patients diagnosed with clinically suspected myocarditis who underwent two consecutive CMR examinations between September 2015 and March 2022 at our institution, no more eligible cases were available. A prospective study with larger sample size is need to verify our findings.
Q2:
It seems that the reduction in the statistical significance can be improved when switching to a narrowed heart rate range. It further seems that there can be many other ways of investigating the relationship between heart rate and strains. For example, IDI, NRI, AIC or BIC analysis comparing models where heart rate is included and excluded, evaluation of correlation strengths of individual parameters in modelling, leave-one-out analysis etc. The authors are urged to consider this.
Weise Valdés et al, Left-Ventricular Reference Myocardial Strain Assessed by Cardiovascular Magnetic Resonance Feature Tracking and fSENC-Impact of Temporal Resolution and Cardiac Muscle Mass
November 2021Frontiers in Cardiovascular Medicine 8:764496
Response: Thanks for the constructive advice. Fast strain-encoded (fSENC) CMR imaging highlights the influence of gender, temporal resolution, cardiac muscle mass and age on myocardial strain values as reported by Weise Valdés et. al.[2] It is of significant importance to investigate the relationship of heart rate and myocardial strains in suspected myocarditis by using fSENC. However, MyoStrain® software is not available in our institution. Further studies are needed to compare FT strains and fSENC strains in suspected myocarditis.
Q3: The methodology is acceptable, but the absence of mapping and ecv should be mentioned in the limitation.
Response: Thanks for the constructive advice, we mentioned it in the study limitations.
Reviewer 3 Report
Yan Chen et al. present an interesting study on atrial and ventricular strain in patients with suspected myocarditis. There are a few methodological weaknesses with regard to the diagnostic certainty of the described collective. However, this is adequately addressed in the article. In my view, some points need further clarification:
1. A major factor in most CMR studies of myocarditis is the definition of the diagnostic gold standard. As correctly described in the methods section, the present work deals with cases of suspected myocarditis. In this respect, the description "acute myocarditis" in the rest of the manuscript is rather misleading. This fact is all the more important as only 18% met the LLC and there are suspected acute myocarditis without troponin. I recommend a more concrete wording here, especially in the title (consequently “suspected myocarditis”). In addition, the term "acute" should be defined more precisely (differentiation from subacute forms, mostly symptom onset less than 2 weeks). A patient without troponin and without LLC is not likely to have myocarditis. A suggestion here would be to use the wording according to the Bonaca criteria, which has become established as a standard for scientific use.
2. The case number of 55 patients is OK for the description of changes in strain, although the high rate of poor image quality is surprising. It should only be stated whether this is a retrospective or a prospective analysis. It seems to be a retrospective image analysis with prospective clinical follow-up. However, assessing MACE in a regression analysis would actually require more cases. Was there a calculation of the needed sample size beforehand?
3. Figure 3: The legibility of the table annotation is poor because the font size is too small.
4. Was there a baseline characteristic or CMR parameter in the initial scan that could also predict MACE? The clinical impact of differences in atrial strain seems rather small to me.
5. Lines 262 to 275: In my view, a transient or acute change in LA function is more likely to be due to a change in LV filling pressures than to contractile dysfunction or inflammation. This point should be included in the discussion. LVEDP is known as a predictive factor in many acute and chronic cardiac diseases and is even a diagnostic criterion (e.g. HFpEF). Even part of the improvement in RV function in terms of RVGLS is due to improvement in LV filling pressures and reduced congestion across the pulmonary stromal bed.
6. Lines 300 to 302: For CMR feature tracking, I assume no influence of the vendor on the strain results. With mapping, of course, it would be different.
Author Response
Yan Chen et al. present an interesting study on atrial and ventricular strain in patients with suspected myocarditis. There are a few methodological weaknesses with regard to the diagnostic certainty of the described collective. However, this is adequately addressed in the article. In my view, some points need further clarification:
- A major factor in most CMR studies of myocarditis is the definition of the diagnostic gold standard. As correctly described in the methods section, the present work deals with cases of suspected myocarditis. In this respect, the description "acute myocarditis" in the rest of the manuscript is rather misleading. This fact is all the more important as only 18% met the LLC and there are suspected acute myocarditis without troponin. I recommend a more concrete wording here, especially in the title (consequently “suspected myocarditis”). In addition, the term "acute" should be defined more precisely (differentiation from subacute forms, mostly symptom onset less than 2 weeks). A patient without troponin and without LLC is not likely to have myocarditis. A suggestion here would be to use the wording according to the Bonaca criteria, which has become established as a standard for scientific use.
Response: Thank you for the valuable advice, we have changed the wording in the revised manuscript.
2. The case number of 55 patients is OK for the description of changes in strain, although the high rate of poor image quality is surprising. It should only be stated whether this is a retrospective or a prospective analysis. It seems to be a retrospective image analysis with prospective clinical follow-up. However, assessing MACE in a regression analysis would actually require more cases. Was there a calculation of the needed sample size beforehand?
Response: This is a retrospective study. The sample size should be at least 44 to expect robust results as calculated in PASS software, thus, we believe it is appropriate to assess MACE in a Cox proportional hazard regression analysis.
3. Figure 3: The legibility of the table annotation is poor because the font size is too small.
Response: We changed the font size to make it more legible.
4. Was there a baseline characteristic or CMR parameter in the initial scan that could also predict MACE? The clinical impact of differences in atrial strain seems rather small to me.
Response: Univariate association of the first laboratory and CMR parameters with MACE was added in the manuscript to replace the previous Table 3, initial LA SRa showed significant association with MACE. We analyzed the LA SRa in multivariable models as shown in Table 4.
5. Lines 262 to 275: In my view, a transient or acute change in LA function is more likely to be due to a change in LV filling pressures than to contractile dysfunction or inflammation. This point should be included in the discussion. LVEDP is known as a predictive factor in many acute and chronic cardiac diseases and is even a diagnostic criterion (e.g. HFpEF). Even part of the improvement in RV function in terms of RVGLS is due to improvement in LV filling pressures and reduced congestion across the pulmonary stromal bed.
Response: The constructive advice was added to the discussion part. In a recent study of a murine model with viral myocarditis [1], transient atrial inflammation was proved pathologically. The change of LA function may be a result of contractile dysfunction and inflammation. A change in LV filling pressures may be part of the cause as well.
6. Lines 300 to 302: For CMR feature tracking, I assume no influence of the vendor on the strain results. With mapping, of course, it would be different.
Response: Thanks for the helpful advice, we have removed it.
Round 2
Reviewer 2 Report
The attached changes have made the work much more understandable.
It is now very interesting for readers.
In the conclusion is an emphasis on the known fact that although a significant improvement in LV and RV strain parameters could be expected, but Initial LV and RV strain parameters were not significantly associated with MACE, while initial LA SRa showed a significant association with MACE, is welcome and makes conclusion stronger.
Reviewer 3 Report
The manuscript by Yan Chen et al. has improved significantly. Two main points about the new manuscript particularly appeal to me: 1. the readability has increased significantly due to the linguistic adjustments and some results are now clearer. 2. the focus on baseline CMR as a prognostic marker has a much greater clinical impact than the difference between baseline and follow-up. Although I wonder how the changed statistics came about, the change is welcome. The prognostic impact of the LA bosster pump strain is of real interest.
The changes made the manuscript coherent. Results and discussion are adequately presented. Methods and limitations are clear and comprehensible.